# Novel Metabolites Associated with Decreased GFR in Finnish Men: A 12-Year Follow-Up of the METSIM Cohort

**DOI:** 10.3390/ijms251810044

**Published:** 2024-09-18

**Authors:** Lilian Fernandes Silva, Jagadish Vangipurapu, Anniina Oravilahti, Markku Laakso

**Affiliations:** 1Institute of Clinical Medicine, Internal Medicine, University of Eastern Finland, 70211 Kuopio, Finland; lilianf@uef.fi (L.F.S.); jagadish.vangipurapu@uef.fi (J.V.); anniina.oravilahti@uef.fi (A.O.); 2Department of Medicine, Division of Cardiology, David Geffen School of Medicine, University of California, Los Angeles, CA 90095, USA; 3A.I. Virtanen Institute for Molecular Sciences, University of Eastern Finland, 70211 Kuopio, Finland; 4Department of Medicine, Kuopio University Hospital, 70200 Kuopio, Finland

**Keywords:** glomerular filtration rate, metabolites, metabolomics, metabolic pathways

## Abstract

Identification of the individuals having impaired kidney function is essential in preventing the complications of this disease. We measured 1009 metabolites at the baseline study in 10,159 Finnish men of the METSIM cohort and associated the metabolites with an estimated glomerular filtration rate (eGFR). A total of 7090 men participated in the 12-year follow-up study. Non-targeted metabolomics profiling was performed at Metabolon, Inc. (Morrisville, NC, USA) on EDTA plasma samples obtained after overnight fasting. We applied liquid chromatography mass spectrometry (LC-MS/MS) to identify the metabolites (the Metabolon DiscoveryHD4 platform). We performed association analyses between the eGFR and metabolites using linear regression adjusted for confounding factors. We found 108 metabolites significantly associated with a decrease in eGFR, and 28 of them were novel, including 12 amino acids, 8 xenobiotics, 5 lipids, 1 nucleotide, 1 peptide, and 1 partially characterized molecule. The most significant associations were with five amino acids, N-acetylmethionine, N-acetylvaline, gamma-carboxyglutamate, 3-methylglutaryl-carnitine, and pro-line. We identified 28 novel metabolites associated with decreased eGFR in the 12-year follow-up study of the METSIM cohort. These findings provide novel insights into the role of metabolites and metabolic pathways involved in the decline of kidney function.

## 1. Introduction

Chronic kidney disease (CKD) affects approximately 10% of the Western countries’ population [1]. Glomerular filtration rate (GFR) is accepted as the best marker of impaired kidney function, calculated as an estimated GFR (eGFR) [2]. Diabetes is a major risk factor for impaired kidney function [3], but also age, sex, hypertension, obesity, increased total triglycerides, and smoking are risk factors for CKD [4]. During the last few years, genome-wide association studies identified hundreds of genetic variants for kidney diseases [5,6,7]. Interestingly, a recent study identified genetic variants in the individuals with and without diabetes and reported that a majority of eGFR loci were similar in individuals with and without diabetes [8]. Genetic studies advanced our understanding of CKD, but they explain only a portion of the disease progression. Additionally, clinical markers often detect an impairment in kidney function only at later stages.

The first studies aiming to identify metabolites associated with eGFR had a small size and included only a low number of metabolites [9,10,11,12,13]. Grams et al. [14] included 587 participants in their study, and identified five metabolites (16-hydroxypalmitate, kynurenate, homovanillate sulphate, N2,N2-dimethylguanosine, and hippurate) associated with CKD. Lin et al. performed a large metabolome-wide association study, including 640 metabolites in 3906 participants of the Hispanic Community Health Study/Study of Latinos. They identified 404 eGFR-metabolite associations and found 79 novel associations [15], where amino acids and xenobiotics were the most frequent metabolites associated with eGFR. Recently, two studies reported several metabolites associated with CKD [16,17].

Identification of novel metabolites offers additional insights into the underlying biochemical processes that are lacking in genetic or clinical markers. Metabolites can serve as early biomarkers and potentially modifiable risk factors. Early identification of individuals having impaired kidney function is essential in the prevention of CKD and its complications. However, previous studies aiming to identify metabolites associated with a decrease in eGFR were cross-sectional and included a small number of participants and metabolites. Furthermore, many of these studies never addressed the long-term progression of CKD, leaving the gaps in the understanding of the changes in metabolites over time. Our population-based study included 10,159 participants having 1009 metabolites measured at baseline. A total of 7090 participated in a 12-year follow-up. Our study is the largest study identifying novel metabolites associated with a decline in eGFR during a long follow-up. Therefore, our study has a good statistical power to reveal new metabolic pathways for impaired kidney function.

## 2. Results

### 2.1. Baseline Characteristics

We included in our study 10,159 METSIM participants. Table 1 shows the baseline characteristics of the participants according to their glucose tolerance. These groups differed significantly in age, systolic blood pressure, BMI, total triglycerides, fasting glucose, HbA1c, fasting plasma insulin, eGFR, urine albumin, and high-sensitivity C-reactive protein (hs-CPR). The difference between the three groups was statistically significant but not clinically relevant eGFR (87.9 in the NGT group, 88.6 and 86.1 in the T2D group).

### 2.2. Metabolites in Participants with Decreased and Normal eGFR

We compared metabolite concentrations between the participants having eGFR < 80 and eGFR ≥ 80 ml/min/1.73 m2 and found statistically significant (*p* < 5 × 10^−5^) differences in 586 metabolites. The top significant 100 associations are shown in Appendix A. The most significant differences (*p* < 1.1 × 10^−350^) between the two groups were in 1-methylhistidine, 1-methyl-4-imidazoleacetate, 2,3-dihydroxy-5-methylthio-4-pentenoate (DMTPA), creatinine, hydroxy asparagine, N,N,N-trimethyl-alanylproline betaine, N-acetylalanine, and pseudouridine.

### 2.3. Effects of Glucose Tolerance on Metabolic Profile

We analyzed the associations of eGFR with metabolites in different subgroups of glucose tolerance (n = 1 057 in each group matched for age and BMI). Participants with NGT had 379 statistically significant associations with the metabolites, participants with prediabetes had 474 significant associations, and participants with T2D had 378 significant associations. Appendix A presents the 100 most significant metabolites in the participants with T2D, prediabetes, and NGT. Independently of the glucose tolerance, all metabolites were associated with a decrease in eGFR. The Venn diagram (Figure 1) shows that the participants in the different glucose tolerance groups shared 78% of the 100 most significant metabolite associations, 11 of the metabolites were found only in the NGT group, 7 in the prediabetes group, and 12 in the T2D group.

### 2.4. Metabolites Associated with a Decrease in eGFR

We performed linear regression to associate 1009 metabolites with eGFR at baseline without adjustment for confounding factors, adjusted for baseline eGFR (Model 1), and adjusted for baseline eGFR, age, BMI, smoking, fasting glucose, total triglycerides, and systolic blood pressure (Model 2) (Appendix A). All metabolites listed in Appendix A had *p* < 5 × 10^−5^ in all models. Adjustment for the baseline eGFR (Model 1) substantially decreased beta and *p* values. In Model 2 beta and *p* value further deceased but the decreases were relatively small.

We found 108 metabolites significantly associated with a decrease in eGFR (Appendix A), and 28 of them were novel (Table 2). The 10 most statistically significant metabolites associated with decreased eGFR were six amino acids, creatinine, hydroxyasparagine, N,N,N-trimethyl-alanylproline betaine, N-acetylalanine, N-acetylserine, C-glycosyltryptophan, and N-formylmethionine; a nucleotide pseudouridine; xenobiotics erythritol; and carbohydrate erythronate.

Among the novel 28 metabolites decreasing eGFR, 12 were amino acids, 5 lipids, 1 nucleotide, 1 peptide, 8 xenobiotics, and 1 a partially characterized molecule. Among the amino acids, the three most significant imverse associations were with N-acetylmethionine (beta = −0.087, *p* = 5.5 × 10^−24^), N-acetylvaline (beta = −0.082, *p* = 2.6 × 10^−21^), and γ-carboxyglutamate (beta = −0.065, *p* = 2.6 × 10^−14^). Among the lipids, the two most significant inverse associations were with 11beta-hydroxyetiocholanolone (beta = −0.050, *p* = 4.0 × 10^−7^), and 2-methylmalonylcarnitive C4-DC (beta = -0.042, *p* = 3.1.0 × 10^−6^), and among the xenobiotics for 2,3-dihydroxyisovalerate (beta = −0.048, *p* = 6.8 × 10^−6^, and (S)-a-amino-omega-caprolactam (beta = −0.050, *p* = 1.0 × 10^−8^).

### 2.5. Genetic Variants Associated with Novel Metabolites 

We identified nine genetic variants significantly associated with the novel metabolites (Table 3). The most significant associations were with 5-methyluridine, glycine, proline, and N-acetylmethionine. Each of the nine genetic variants were associated with at least three different metabolites, suggesting pleiotropy of these genes. Importantly, none of these genetic variants were significantly associated with a decrease in eGFR, indicating that the effects of the metabolites on eGFR were not explained by genetic factors.

## 3. Discussion

We measured 1009 metabolites with LC-MS/MS in 10,188 participants of the METSIM study. Our study reports several novel findings. We found that glucose tolerance did not have a major effect on the metabolite profile at baseline. Among the top 100 metabolites associated with eGFR, 78 were identical in participants with normal glucose tolerance, pre-diabetes, and diabetes (Figure 1). Our results suggest that the metabolic pathways leading to a decrease in eGFR are largely independent of glucose tolerance. This observation agrees with a previous study reporting that the majority of the eGFR loci were similar in the individuals with and without diabetes [8].

We found several statistically significant associations of the metabolites with a decrease in eGFR in the 12-year follow-up of the METSIM cohort. Of the 108 metabolites associated with a decrease in eGFR, 28 were novel (Table 2). We also replicated metabolite associations with decreased eGFR reported in previous studies [13,18,19,20,21,22,23,24,25,26]. The metabolic pathways independent from glucose highlights the complexity in the progression of CKD. Our findings suggest that non-glucose pathways, including amino acids, lipids, and xenobiotics, have independent effects on kidney function. However, our findings do not change the current treatment of patients with diabetes having impaired kidney function. Medication lowering hyperglycaemia remains the primary treatment in patients with diabetes and CKD. 

We found three novel associations of N-acetylated amino acids (N-acetylmethionine, N-acetylvaline, and N-acetyltaurine) with a decrease in eGFR. N-acetylated amino acids are uremic toxins [27]. Aminoacylase-1 (ACY1) enzyme converts acetylated amino acids into free amino acids, and therefore the individuals having impaired activity of ACY1 or a mutation in the ACYL1 gene have increased concentrations of acetylated amino acids in blood and urine [28,29,30,31,32,33]. N-acetylated amino acids are uremic toxins that accumulate in the blood due to impaired kidney function (27). These toxins disrupt cellular processes, promote inflammation, and induce oxidative stress, worsening CKD progression (27–32). This highlights the potential of the metabolites to identify therapeutic targets for the prevention of CKD progression.

Amino acid γ-carboxyglutamate was significantly associated with a decrease in eGFR. γ-carboxyglutamate is a calcification inhibitor [34]. Atherosclerotic and vascular calcification are closely linked to the vitamin K-dependent protein matrix γ-carboxyglutamate. Vitamin K antagonists, including warfarin, are associated with increased calcification of renal and other arteries [34,35]. Coronary artery calcification was previously associated with a decline in eGFR [36]. Increased levels of γ-carboxyglutamate may represent a compensatory response to counteract vascular calcification as kidney function declines. 

We report three novel associations of N-lactoyl-amino acids (N-lactoylvaline, N-lactoylisoleucine, and N-lactoylphenylalanine) with a decrease in eGFR. N-lactoylphenylalanine concentrations are increased in patients with phenylketonuria [37]. These patients have increased oxidative stress leading to tubulointerstitial disease, impaired kidney function, proteinuria, and arterial hypertension [38,39]. N-lactoylvaline and N-lactoylisoleucine were found in the urine of a patient with maple syrup urine disease [40], which is associated with nephrotic syndrome [41].

We also found that the nucleoside 5-methyluridine (ribothymidine), an endogenous methylated nucleoside, decreased eGFR. This finding was previously reported in rats with CKD [42]. Altered DNA methylation modulates the expression of pro-inflammatory and pro-fibrotic genes, stimulating renal disease progression [43]. High concentrations of homocysteine, hypoxia, and inflammation alter the epigenetic regulation of gene expression in CKD, impacting eGFR [43]. 

Eight of the 28 novel metabolites impairing eGFR were xenobiotics, chemical substances within an organism that are not naturally produced. Xenobiotics are food components, plant constituents, pesticides, industrial chemicals, environmental pollutants, or benzoate metabolites. An organic compound (S)-a-amino-omega-caprolactam is a uremic solute previously shown to impair kidney function [44]. 3-methyl catechol sulfate, a marker of current smoking and coffee consumption [45], decreased eGFR in our study. We also showed that genetic variants were not associated with xenobiotics, suggesting that decreased eGFR is largely regulated also by lifestyle and environmental factors.

Our findings highlight multiple metabolic pathways associated with a decrease in eGFR. We identified 28 novel metabolites among amino acids, lipids, nucleotides, peptides, and xenobiotics associated with decreased eGFR. Eight xenobiotics were associated with a decrease in eGFR, showing that non-genetic factors, including benzoate pathway, food components, and plants play a significant role in kidney dysfunction, demonstrating the influence of environmental factors on eGFR. Additionally, the effects of N-lactoyl-amino acids and 5-methyluridine show a potential for epigenetic regulation of kidney function. Overall, our novel findings provide valuable insights into the complex biochemical interactions affecting kidney function and pave the way for future studies to explore metabolic pathways on kidney function in diverse populations.

The strength of our study is that the METSIM study is the largest randomly selected population-based cohort identifying metabolites associated with a decrease in eGFR by applying the LC-MS/MS analysis method. Additionally, we followed our cohort for 12 years, and at baseline and follow-up, the metabolites identified were inversely associated with eGFR, increasing the credibility of our findings. We applied a conservative statistical significance threshold in all analyses to obtain reliable conclusions. 

Our study has several limitations. The homogeneity of the study population (middle-aged and elderly Finnish men) limits the generalizability of our findings. Therefore, the replication of our findings in more diverse populations, such as women and non-European cohorts, is needed. Our study was an observational study, and therefore, we cannot establish causality between the metabolites and the decline of kidney function. Additionally, there may be unmeasured confounders, such as medication use or environmental factors, having effects on our findings. The LC-MS/MS platform provides a broad metabolite coverage, but the results are not fully generalizable to other metabolomics platforms.

In vitro and animal studies could help to establish causal relationships between the metabolites and the decline of the kidney function. Incorporating these findings into CKD risk prediction models may improve early detection and personalized treatment. Additionally, targeting specific metabolic pathways, for example, those involving uremic toxins, could reveal novel therapeutic approaches. Expanding the approach to include other omics data, such as genomics and proteomics, could further enhance our understanding of CKD progression.

## 4. Materials and Methods

### 4.1. Study Population and Laboratory Measurements 

The METabolic Syndrome in Men (METSIM) study includes 10,197 men, aged from 45 to 73 years at baseline, and randomly selected from the population register of Kuopio, Eastern Finland. The METSIM study was approved by the Ethics Committee of the Kuopio University Hospital, Finland. All participants provided written informed consent. 

The design and methods of the METSIM study were previously described in detail [46,47]. A total of 10,159 men were included in the current study, 3034 had normal glucose tolerance (NGT, fasting glucose < 6.1 mmol/L, 2-hour glucose < 7.8 mmol/L), 5715 prediabetes [impaired fasting glucose (6.1–6.9 mmol/L) or impaired glucose tolerance (7.8 to 11.0 mmol/L) or both], and 1410 T2D, [fasting glucose ≥ 7.0 mmol/L, or 2-hour glucose ≥ 11.1 mmol/L or glycated hemoglobin A1c (HbA1c) ≥ 6.5%] according to the American Diabetes Association classification [48]. BMI was calculated as weight divided by height squared. Smoking status was defined as current smoking (yes/no). All participants, excluding participants with T2D at baseline, underwent a 2-hour oral glucose tolerance test (75 g of glucose), and samples for plasma glucose and insulin were drawn at 0, 30, and 120 min. Other laboratory measurements were previously explained [46]. eGFR was calculated using the CKD-Epi equation [49].

### 4.2. Metabolomics

Non-targeted metabolomics profiling was performed at Metabolon, Inc. (Morrisville, NC, USA) on EDTA plasma samples obtained after overnight fasting, as previously described in detail [47,50]. We applied liquid chromatography mass spectrometry (LC-MS/MS) to identify the metabolites (the Metabolon DiscoveryHD4 platform). The LC-MS/MS platform was chosen for its high sensitivity, specificity, and broad dynamic range, making it ideal for detecting a wide variety of metabolites. Compared to other platforms, especially proton NMR, LC-MS/MS offers superior sensitivity, allowing for the identification of subtle metabolic changes, which is crucial in discovering early biomarkers for kidney function decline. Although limitations such as ion suppression and complex data processing exist for LC-MS/MS, its advantages in sensitivity and metabolite coverage makes it the best choice for this study. All samples were processed together for peak quantification and data scaling. We quantified raw mass spectrometry peaks for each metabolite using the area under the curve, and evaluated overall process variability by the median relative standard deviation for endogenous metabolites present in all 20 technical replicates in each batch. We adjusted for variation caused by day-to-day instrument tuning differences and columns used for biochemical extraction by scaling the raw peak quantifications to the median for each metabolite by the Metabolon batch.

### 4.3. Selection of Genetic Variants Decreasing Glomerular Filtration Rate

We identified genetic variants associated with a decrease in eGFR from previously published studies and the GWAS Catalog (The NHGRI-EBI Catalog of human genome-wide association studies (https://www.ebi.ac.uk/gwas/—accessed on 4 July 2024) in individuals of European ancestry. Altogether, 117 genes were found to be associated with impaired eGFR. 

### 4.4. Statistical Analysis

We conducted statistical analyses using IBM SPSS Statistics, version 29. We log-transformed all continuous variables except for age and follow-up time to correct for their skewed distribution. We performed association analyses between the eGFR and metabolites using linear regression adjusted for confounding factors (Model 1, adjustment for eGFR at baseline, Model 2, adjustment for eGFR at baseline, age, BMI, smoking, systolic blood pressure, fasting glucose, and total triglycerides). The variables in Model 2 (age, BMI, smoking, systolic blood pressure, fasting glucose, and triglycerides) were selected because they are well-known risk factors for both CKD progression and metabolic changes. These variables were chosen to reduce bias and ensure that the associations between metabolites and eGFR are not influenced by these factors. We give the results as standardized beta coefficients and *p* values with the metabolite as a dependent variable. We used one-way ANOVA to assess the differences in clinical traits and metabolites between the two groups at baseline. We applied the Bonferroni correction to determine statistical significance for the metabolites identified (*p* < 5.0 × 10^−5^).

## 5. Conclusions

We measured 1009 metabolites in 10,159 Finnish men of the METSIM cohort and associated the metabolites with eGFR in the 12-year follow-up study. We found 108 metabolites significantly associated with a decrease in eGFR, and 28 of them were novel, including especially amino acids, xenobiotics, and lipids, showing that hyperglycaemia is not the only cause for impaired eGFR. Our findings provide novel insights into the role of metabolites and metabolic pathways involved in the decline of kidney function.

## Figures and Tables

**Figure 1 ijms-25-10044-f001:**
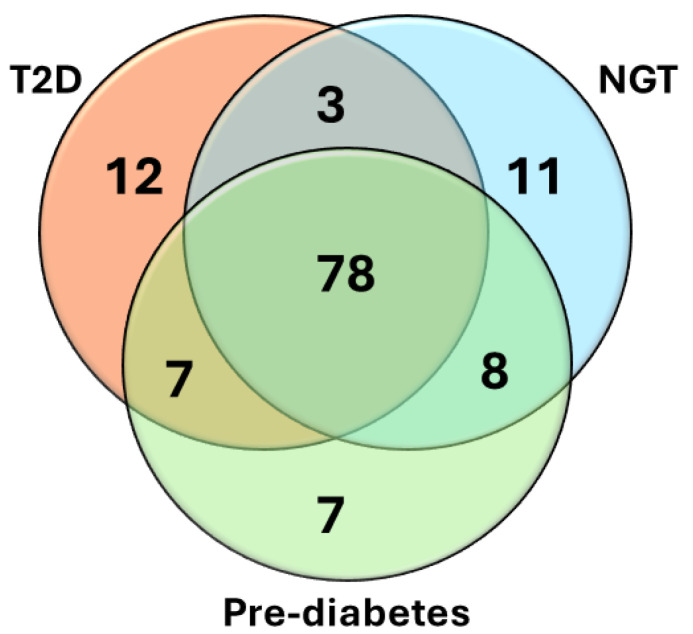
Venn diagram shows the top 100 most significant metabolites associated with eGFR across the participants with normal glucose tolerance (NGT), prediabetes, and type 2 diabetes (T2D).

**Table 1 ijms-25-10044-t001:** Baseline Characteristics of the Participants According to Glucose Tolerance.

Measurements	NGT (n = 3034)	Pre-Diabetes (n = 5715)	T2D (n = 1410)	*p*
Age (years)	56.8 ± 6.9	57.4 ± 7.2	60.6 ± 6.7	1.1 × 10^−63^
Systolic blood pressure (mmHg)	134.3 ± 15.9	138.7 ± 16.2	145.2 ± 18.1	2.1 × 10^−93^
Body mass index (kg/m^2^)	25.8 ± 3.38	27.4 ± 3.9	30.2 ± 5.2	1.1 × 10^−247^
Current smoking (%)	18.0	18.4	17.2	0.606
Total triglycerides (mmol/l)	1.22 ± 0.65	1.49 ± 1.08	1.90 ± 1.21	1.2 × 10^−143^
Fasting glucose (mmol/l)	5.24 ± 0.24	5.97 ± 0.37	7.51 ± 2.01	<1 × 10^−250^
HbA1C (%)	5.59 ± 0.31	5.71 ± 0.34	6.58 ± 1.13	<1 × 10^−250^
Fasting plasma insulin (mU/l)	6.25 ± 4.11	9.32 ± 6.4	19.6 ± 28.5	<1 × 10^−250^
Creatinine (umol/l)	84.6 ± 15.9	83.4 ± 12.8	84. 6 ± 22.3	0.0003
eGFR (ml/min/1.73 m^2^)	87.9 ± 12.3	88.6 ± 12.2	86.1 ± 14.5	4.5 × 10^−10^
Urine albumin (mg/l)	18.4 ± 110.9	20.6 ± 82.5	93.5 ± 380.1	7.2 × 10^−181^
hs-CRP (mg/l)	1.82 ± 2.96	2.13 ± 4.5	3.22 ± 6.07	3.4 × 10^−40^

Abbreviations: NGT, normal glucose tolerance; T2D, type 2 diabetes; HbA1C, hemoglobin A1C; eGFR, estimated glomerular filtration rate; and hs-CRP, high sensitivity C-reactive protein.

**Table 2 ijms-25-10044-t002:** Novel Metabolites Associated with a Decrease In eGFR.

Metabolite	Sub-Class	N	Beta	*p* *	Beta	*p* **
Amino acids						
N-acetylmethionine	Methionine, cysteine, taurine metabolism	7080	−0.334	1.4 × 10^−183^	−0.087	5.5 × 10^−24^
N-acetylvaline	Leucine, isoleucine, valine metabolism	7082	−0.343	1.0 × 10^−194^	−0.082	2.6 × 10^−21^
γ-carboxyglutamate	Glutamate metabolism	6929	−0.295	1.1 × 10^−138^	−0.065	2.6 × 10^−14^
3-methylglutaryl- carnitine (2)	Leucine, isoleucine, valine metabolism	7001	−0.257	1.1 × 10^−105^	−0.058	5.8 × 10^−12^
Proline	Urea cycle; arginine proline metabolism.	7081	−0.107	1.3 × 10^−19^	−0.048	3.9 × 10^−9^
Pro-hydroxy-pro	Urea cycle; arginine proline metabolism	7079	−0.155	1.9 × 10^−39^	−0.047	5.2 × 10^−9^
4-guanidinobutanoate	Guanidino acetamido metabolism	7049	−0.158	1.7 × 10^−40^	−0.049	2.3 × 10^−9^
N-acetyltaurine	Methionine, cysteine, taurine metabolism	7048	−0.208	1.4 × 10^−69^	−0.041	7.6 × 10^−7^
Hydantoin-5-propionate	Histidine metabolism	6154	−0.211	3.6 × 10^−63^	−0.043	1.1 × 10^−6^
N-lactoylvaline	Lactoyl amino acid	6781	−0.182	2.5 × 10^−51^	−0.043	3.1 × 10^−6^
N-lactoylisoleucine	Lactoyl amino acid	5437	−0.189	4.4 × 10^−45^	−0.043	1.6 × 10^−5^
N-lactoylphenylalanine	Lactoyl amino acid	7033	−0.233	2.7 × 10^−87^	−0.037	4.4 × 10^−5^
Lipids						
11beta-hydroxy etiocholanolone glucuronide	Androgenic steroids	4891	−0.204	2.9 × 10^−47^	−0.050	4.0 × 10^−7^
3-decenoylcarnitine	Fatty acid metabolism	5395	−0.217	2.9 × 10^−58^	−0.042	9.2 × 10^−6^
Cis-3,4-methylene heptanoylglycine	Fatty acid metabolism	6825	−0.161	5.2 × 10^−41^	−0.038	4.8 × 10^−6^
2-methylmalonyl carnitine (C4-DC)	Fatty acid metabolism	5827	−0.235	8.0 × 10^−74^	−0.042	3.1 × 10^−6^
Propionylglycine	Fatty acid metabolism	3960	−0.119	4.9 × 10^−14^	−0.049	1.3 × 10^−5^
Nucleotide						
5-methyluridine(ribothymidine)	Pyrimidine metabolism	7082	−0.134	6.8 × 10^−30^	−0.038	3.1 × 10^−6^
Peptide						
Pyroglutamylvaline	Modified peptides	6398	−0.202	7.7E × 10^−60^	−0.051	2.6 × 10^−9^
Xenobiotics						
2,3-dihydroxyisovalerate	Food component/plant	6998	−0.206	3.8 × 10^−68^	−0.048	6.8 × 10^−9^
(S)-a-amino-omega-caprolactam	Food component/plant	7007	−0.296	1.3 × 10^−141^	−0.050	1.0 × 10^−8^
3-methoxycatechol sulfate (2)	Benzoate metabolism	5379	−0.185	2.0 × 10^−42^	−0.044	1.9 × 10^−6^
3-methyl catechol sulfate (1)	Benzoate metabolism	7065	−0.209	3.0 × 10^−70^	−0.040	2.1 × 10^−6^
3-methoxycatechol sulfate (1)	Benzoate metabolism	6318	−0.174	4.0 × 10^−44^	−0.039	5.5 × 10^−6^
2-acetamidophenol sulfate	Food component/plant	5939	−0.153	2.9 × 10^−32^	−0.042	3.6 × 10^−6^
N-(2-furoyl)glycine	Food component/plant	5025	−0.235	5.0 × 10^−64^	−0.042	2.4 × 10^−5^
2-aminophenol sulfate	Food component/plant	7066	−0.147	2.8 × 10^−35^	−0.036	1.1 × 10^−5^
Other metabolites						
Glutamine_degradant	Partially characterized molecules	7060	−0.222	7.3 × 10^−80^	−0.071	2.2 × 10^−17^

*p* *: non-adjusted; *p* **: adjusted for eGFR at baseline, age, BMI, smoking, fasting glucose, total triglycerides, and systolic blood pressure.

**Table 3 ijms-25-10044-t003:** The Variants of Nine Genes were Associated with the Novel Metabolites Related to a Decline in eGFR.

Gene-Variant	Metabolite	*p*
*KLHDC7B*-rs470118	5-methyluridine	9.9 × 10^−199^
*CPS1*-rs715	Glycine	8.1 × 10^−90^
*AC007326.4*-rs5992344	Proline	2.0 × 10^−63^
*DOCK3*-rs138144932	N-acetylmethionine	1.3 × 10^−44^
*AOX1*-rs7562507	Hydantoin-5-propionate	1.4 × 10^−17^
*COLEC10*-rs13264172	Pro-hydroxy-pro	3.5 × 10^−10^
*MAGI1*-rs264676	2.3-dihydroxy-5-methylthio-4-penenoate	2.9 × 10^−8^
*DCBLD2*-rs192423025	Pyroglutamylvaline	3.4 × 10^−8^
*CNTNAP2*-rs533473709	γ-carboxyglutamate	5.3 × 10^−8^

## Data Availability

The data that support the findings of this study are available from the corresponding author M.L. upon reasonable request.

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
