# Peer review of "Novel Metabolites Associated with Decreased GFR in Finnish Men: A 12-Year Follow-Up of the METSIM Cohort"

_ijms, 2024, doi:10.3390/ijms251810044_

Round 1
Reviewer 1 Report
Comments and Suggestions for Authors
Dear authors please rearrange the article: Results are before material and methods and the article ends with statistical analysis and not with conclusions. Also the discussion part is before material and methods. Conclusion part is missing.
I could not acces table s1.
Why did you choose cut off for GFR=80 ml/min/1.73m2?
Reviewer 2 Report
Comments and Suggestions for Authors
The paper presents an interesting study on the identification of novel metabolites associated with a decrease in eGFR over a 12-year period in a large cohort. While the study has several strengths, such as a large sample size and long follow-up duration, there are notable areas for improvement in terms of clarity, methodological rigor, and interpretation of results.
Title and Abstract:
The title accurately reflects the content of the paper. However, it could be more specific about the population studied (e.g., Finnish men).
The abstract provides a good summary but lacks details on the main findings' significance and potential implications. Consider adding a sentence on the broader impact of these findings in clinical or biological terms.
Introduction:
The introduction effectively sets the stage for the study but lacks depth in explaining why novel metabolite discovery is critical for understanding CKD progression beyond current genetic and clinical markers. Expand on this to better justify the study's novelty and importance.The reference to previous studies is helpful, but it would be beneficial to explicitly mention gaps in these studies that your research aims to fill.
Methods:
Population and Study Design: The choice of the METSIM cohort is well-justified. However, more detail on participant selection and exclusion criteria is needed to assess potential biases.
Metabolomics Analysis: There is a lack of clarity regarding the choice of the LC-MS/MS platform and its specific advantages. Why was this platform chosen over others? Were there any limitations?
Statistical Analysis: The statistical methods are broadly appropriate, but the rationale behind certain choices (e.g., the specific confounders adjusted for in Model 2) needs more explanation. Why were these variables chosen, and were any other potential confounders considered?
Results:
Baseline Characteristics: The baseline characteristics table (Table 1) is comprehensive. However, the small differences in eGFR between groups should be better contextualized; why are these differences clinically significant?
Metabolite Associations: The finding of 108 metabolites associated with decreased eGFR is intriguing. However, the presentation is somewhat scattered. Consider reorganizing this section to clearly differentiate between known and novel findings, perhaps using subheadings or a summary table.
Novel Metabolites: The identification of 28 novel metabolites is a key strength, but the discussion around these metabolites lacks depth. What is the biological significance of these novel associations? For instance, the role of N-acetylated amino acids as uremic toxins is mentioned, but this could be expanded upon to discuss potential mechanisms.
Discussion:
The discussion provides a reasonable interpretation of the findings but often lacks critical analysis. For example, while the independence of metabolic pathways from glucose tolerance is highlighted, there is little discussion on how this might impact current understanding or management of CKD. More emphasis should be placed on the potential translational implications of these findings.
Limitations:
The limitations section acknowledges the homogeneity of the study population (middle-aged and elderly Finnish men), which is a significant limitation. However, other potential limitations (e.g., the potential for unmeasured confounding, the observational nature of the study, or the generalizability of the metabolomics platform) should be discussed.
Future Directions: The paper would benefit from a more forward-looking perspective. What are the next steps in this research area? How might these findings be validated or used in future studies?
Figures and Tables:
The Venn diagram in Figure 1 is informative but could be more visually distinct (e.g., use color-coding or shading). Additionally, consider including more descriptive figure legends to aid interpretation.
The supplementary tables are extensive, but navigating them is difficult without a better structure or more guidance within the main text.
Minor Comments
Clarity and Language: There are several instances of awkward phrasing and minor grammatical errors throughout the paper. A thorough proofreading would help improve readability.
References: Ensure that all references are up-to-date and relevant. Some citations are relatively old and may not reflect the current state of research in this field.
Comments on the Quality of English Language
Minor editing is necessary.
Round 2
Reviewer 2 Report
Comments and Suggestions for Authors
Paper can be accepted in its present form.